# Using 311 data to develop an algorithm to identify urban blight for public health improvement

**Jessica Athens**[1]\*, **Setu Mehta**[2], **Sophie Wheelock**[1], **Nupur Chaudhury**[1], **Mark Zezza**[1]

**1** New York State Health Foundation, New York, New York, United States of America, **2** Harvard College, Harvard University, Cambridge, Massachusetts, United States of America

\* athens@nyshealth.org

## Abstract

The growth of administrative data made available publicly, often in near-real time, offers new opportunities for monitoring conditions that impact community health. Urban blight—manifestations of adverse social processes in the urban environment, including physical disorder, decay, and loss of anchor institutions—comprises many conditions considered to negatively affect the health of communities. However, measurement strategies for urban blight have been complicated by lack of uniform data, often requiring expensive street audits or the use of proxy measures that cannot represent the multifaceted nature of blight. This paper evaluates how publicly available data from New York City's 311-call system can be used in a natural language processing approach to represent urban blight across the city with greater geographic and temporal precision. We found that our urban blight algorithm, which includes counts of keywords ('tokens'), resulted in sensitivity ~90% and specificity between 55% and 76%, depending on other covariates in the model. The percent of 311 calls that were 'blight related' at the census tract level were correlated with the most common proxy measure for blight: short, medium, and long-term vacancy rates for commercial and residential buildings. We found the strongest association with long-term (>1 year) commercial vacancies (Pearson's correlation coefficient = 0.16, p < 0.001). Our findings indicate the need of further validation, as well as testing algorithms that disambiguate the different facets of urban blight. These facets include physical disorder (e.g., litter, overgrown lawns, or graffiti) and decay (e.g., vacant or abandoned lots or sidewalks in disrepair) that are manifestations of social processes such as (loss of) neighborhood cohesion, social control, collective efficacy, and anchor institutions. More refined measures of urban blight would allow for better targeted remediation efforts and improved community health.

## Introduction

Public health concerns have contributed to key urban planning strategies since the late 19[th] century, where reforms in sanitation, the introduction of zoning, and land use regulation were a means of acknowledging the health risks of exposure to contaminated air and water [1,2].

**Data Availability Statement:** Data for New York City 311 Service Requests from 2010 to Present are available for download at https://data.cityofnewyork.us/Social-Services/311-Service-

Requests-from-2010-to-Present/erm2-nwe9. Data were loaded into RStudio (version 3.5.1) via the RSocrata library using the following code. # Install RSocrata library library(RSocrata) register_google (key="[include user-specific key]") # Call in NYC data for 01-01-2018 through 06-30-2018 nyc <- as.data.frame(read.socrata("https://data.cityofnewyork.us/resource/fhrw-4uyv.json?$where=created_date between '2018-01-01T12:00:00' and '2018-06-30T23:59:59'")).

**Funding:** The authors received no specific funding for this work.

**Competing interests:** The authors have declared that no competing interests exist.

Such planning tools were also intended to address social ills, as residential overcrowding and limited access to green space were considered to be risks to psychological and 'moral' well-being [2,3,4]. The changes sought were specifically targeted to the *built environment*, best summarized as '. . . all of the physical parts of where we live and work (e.g., homes, buildings, streets, open spaces, and infrastructure)' [3].

The prominence of the built environment as a determinant of population health rose again in the late twentieth and early twenty-first centuries, with significant research outlining its impacts on chronic disease, mental health, and injuries [4]. Starting in the mid-twentieth century, the concept of 'urban blight' emerged to reflect physical, economic, and social decline of neighborhoods [5]. For the purpose of this research, urban blight is considered as physical disorder (e.g., litter, overgrown lawns, or graffiti) and decay (e.g., abandoned lots or sidewalks in disrepair) that are manifestations of social processes such as (loss of) neighborhood cohesion, social control, collective efficacy, and anchor institutions. Whereas many aspects of urban planning and the built environment (such as street grids, capital investments in utilities and other infrastructure), and development of housing stock are either fixed or could only be changed with long-term interventions, the physical decay and disorder that comprise urban blight can be addressed over relatively short time periods.

Standard approaches to measuring blight with secondary data sets include measures of physical decay, such as vacant commercial property, vacant housing units, and vacant lots within a neighborhood [6,7]. Other researchers have looked at measures of physical disorder, otherwise considered neighborhood quality measures, such as mown lawns, litter and debris, delinquent vehicles, and presence/absence of graffiti [6,8]. These quality-related indicators are more difficult to capture, often requiring street audits in person or virtually, using Google Street View [9,10]. Finally, given that blight is considered a physical manifestation of poor social cohesion, a third dimension of blight includes social and economic investment, which can be evaluated through measures such as perceptions of safety, presence of anchor institutions, and community organizing efforts [11].

The body of literature linking urban blight to community health is constrained due to differing definitions of blight and the availability of secondary data sources for operationalizing that definition. As outlined in Maghelal et al., measures such as vacancy rates or tax delinquency rates assess only one facet of blight, whereas other measures—including income, single-parent households, or racial/ethnic composition—are used to proxy for urban blight [12,5]. These proxies are problematic because they rely on correlations between social disadvantage and physical environment characteristics but overlook the systemic causes of both.

The association between built environment features and health behaviors, biomarkers indicative of chronic disease, and health outcomes has been well documented [12–16]. In particular, quality of the built environment—including tree cover and green space, park amenities, sidewalk coverage and maintenance, and presence/absence of environmental toxins (air pollution, lead and other heavy metals in the soil)—are associated with community health. Though the literature specific to urban blight and public health outcomes is less developed, there is strong evidence that urban blight is specifically associated with higher violent crime and gun crime [5], poor mental health [17,18], and even adverse pregnancy outcomes [19]. Experimental research on the causal relationships between urban blight/blight remediation (specifically urban greening efforts) and (1) biomarkers and (2) mental health demonstrates significant improvements on heart rate and depression measures [18, 20]. These improvements were evident in both high- and low-poverty census tracts.

The municipal 311 data systems that started first in Baltimore, MD (1997) and were introduced to New York City in 2003, could serve as a source of information that addresses the paucity of secondary data for a uniform evaluation of urban blight across a municipality [21]. The

311 data system is designed to log non-urgent calls to municipal agencies regarding anything from requests for information on municipal services and benefits enrollment, to reporting a housing violation or making a noise complaint. These data are updated daily, and include geo-location (latitude and longitude), responsible agency, category of complaint, and free text description of the call. Given the volume of data (~3 million calls annually in New York City), and the noise inherent in the data, 311 data analysis requires a 'big data' approach. The free text nature of call descriptions suggests applying natural language processing to identify key words or strings predictive of urban blight.

## Objectives

This research effort aims to evaluate how regularly updated administrative data, namely the 311-call system data, can be used to represent urban blight at fine-grained geographies and, to a more modest extent, time periods. We apply a natural language processing approach to develop an algorithm to identify urban blight-related calls in the 311 data system and explore their distribution across New York City. These results are compared to American Community Survey residential vacancy data and HUD USPS vacancy rates for commercial and residential buildings. These indicators are commonly used proxies for assessing urban blight [5, 17]. An advantage of a 311-based measure of blight is that it provides finer-grained geographic and temporal data. Although not fully developed in this analysis, the 311 data has the potential to disambiguate different types of blight (i.e., decay, disorder, and social/economic investment). Having more refined measures of blight could also allow for better targeted blight remediation efforts. Future work will explore the relationship between a finalized measure of urban blight drawn from 311 calls and community health conditions.

## Data and methods

### Data

New York City 311 Service Requests were the primary data source for algorithm development. Six months of data (January 1, 2018–June 30, 2018), comprising 1.3 million records, were pulled from New York City's Open Data Portal directly into RStudio (v. 3.5.1) using the 'RSo-crata' library [22–24]. Supplemental data on census tract-level population, residential vacan-cies, and median home value were drawn from the American Community Survey (ACS) 5-Year Estimates, 2013–2017, table DP04 [25]. As with 311 call data, these data were called into RStudio with the 'ACS' library [26]. Data on short- and long-term residential and com-mercial vacancies originated from the Residential and Commercial Vacancy data set from the US Postal Service and Department of Housing and Urban Development for the first quarter of 2018 [27].

### Methods

Following standard practice for Natural Language Processing (NLP), we followed a four-step process: (1) data training, (2) cleaning and tokenization, (3) classification, and (4) validation [28].

Training the data requires the manual designation of a random sample of calls to an urban blight versus a non-urban blight category. We established seven domains of urban blight based on extant literature to guide our data training: social conditions, abandoned property, air qual-ity, street/sidewalk maintenance, noise, sanitary conditions, and building safety. Coding any call into one of these categories signifies the call is urban blight-related. Domains were assigned based on complaint types and free-text call descriptions in the 311 data. Complaint

type is a variable designated by the City of New York and included as part of the 311 system data, which comprises 236 complaint types. We focused on 'high frequency' complaints ($\geq$ 1,000 records) to simplify the training process. The high frequency list included 93 complaint types.

We selected half of the call records (~650,000) as the training data set using simple random sampling. Two raters reviewed a 10% sample of data to assign complaint types (N = 93) to one of the urban blight domains (N = 7) or to a non-urban blight-related category. Consistency in coding was evaluated using Cohen's Kappa statistic, which adjusts for the probability that raters will agree by chance.

After data training, we cleaned the call description text field to address misspelling and to standardize text formatting. String variables were then separated into 'tokens' or unique words or strings that comprise text data. Common tokens such as articles or prepositions were omitted from analysis. Of the remaining tokens, we calculated the percent that appeared exclusively in 'blight' calls and those that were unique to 'non-blight' calls. Ultimately, only tokens that appeared in urban blight-related calls were used in the following stage of analysis.

For data classification, we used a logistic regression model on a 50% sample of the trained data (~325,000 records) to determine how effective the blight-related tokens are at identifying a blight-related call. A token or series of tokens can be used as predictors in the regression model. In order to have parsimonious models, we calculated the total of blight-related tokens that appeared in each record and used this count as the primary predictor of urban blight (0/1) in our logistic regressions.

Our first model was a basic model predicting urban blight, with the unique blight-related token count as the only independent variable (Eq 1). In the second model, we included a categorical variable for borough as an independent variable (Eq 2). Finally, our third model included the variable for borough as well as a variable for the agency assigned responsibility for the 311 call. As with borough, agency was coded as a categorical variable with 15 levels (Eq 3). Responsible agencies included departments of sanitation, police, finance, health and mental hygiene, and consumer affairs, among others (See Table 3 for a full list of agencies).

$$logit(y_i) = \beta_0 + \beta_1(unique\ token\ count_i) + \varepsilon_i \qquad \text{Eq 1}$$

$$logit(y_i) = \beta_0 + \beta_1(unique\ token\ count_i) + \beta_2(borough_i) + \varepsilon_i \qquad \text{Eq 2}$$

$$logit(y_i) = \beta_0 + \beta_1(unique\ token\ count_i) + \beta_2(borough_i) + \beta_3(agency_i) + \varepsilon_i \qquad \text{Eq 3}$$

Urban blight ($y_i$) is a binary variable indicating whether a call is considered blight-related. "Unique token count" represents the number of unique keywords related to blight that appear in any text field within the 311 data. These keywords were those identified as urban related during the data training step. "Borough" is a 5-level factor variable representing Bronx, Brooklyn, Manhattan, Queens, and Staten Island. Bronx serves as the reference category. "Agency" is the agency assigned responsibility for addressing the 311 call. The reference category is unassigned.

The coefficients from each of these three models were used to predict the probability that a call was urban blight-related for the balance of the training data (i.e. categorized data not used in regression models). We used confusion matrices to evaluate how well each model predicted whether a call represented urban blight, calculating accuracy (ACC), positive predictive values (PPV), and negative predictive values (NPV) (Table 1).

As a first effort to validate our results, we calculated correlations between census tract-level measures of housing vacancies from the American Community Survey and blight-related calls,

**Table 1. Sensitivity, specificity, and accuracy of urban blight algorithm.**

| Model 1 | | | | |
|---|---|---|---|---|
| | Urban blight = 1 | Urban blight = 0 | PPV (Sensitivity) | 91% |
| Pred(Urban blight = 1) | 210,413 | 43,701 | NPV (Specificity) | 55% |
| Pred(Urban blight = 0) | 20,718 | 54,211 | ACC (Accuracy) | 80% |
| Model 2 | | | | |
| | Urban blight = 1 | Urban blight = 0 | PPV (Sensitivity) | 90% |
| Pred(Urban blight = 1) | 208,947 | 42,172 | NPV (Specificity) | 57% |
| Pred(Urban blight = 0) | 22,184 | 55,740 | ACC (Accuracy) | 80% |
| Model 3 | | | | |
| | Urban blight = 1 | Urban blight = 0 | PPV (Sensitivity) | 90% |
| Pred(Urban blight = 1) | 208,213 | 23,884 | NPV (Specificity) | 76% |
| Pred(Urban blight = 0) | 22,918 | 74,028 | ACC (Accuracy) | 86% |

both as count variables and expressed as percentages. Similarly, we calculated correlations between blight-related calls (%) with percent of residential and commercial addresses considered long (>12 months), medium (6–12 months), and short-term (< 6 months) vacancies from the USPS/HUD data set.

## Results

Of the 1.3 million call records across 236 complaint types, we identified 93 'high frequency' types ($\geq$ 1,000 records) that comprised 98% of all calls over the 6-month time frame (Table 2). After parallel coding of a sample of the data, in which raters assigned a call to one of the seven urban blight domains (social conditions, abandoned property, air quality, street/sidewalk maintenance, noise, sanitary conditions, building safety) or 'not urban blight related,' we calculated Cohen's Kappa statistic for categorical outcomes [29]. The resulting value, $\kappa = 0.81$, indicated high values of agreement between raters.

The next stage, text cleaning and tokenization, resulted in 1,113 unique words (tokens) that were represented in the call description field. Of these, 46% (516) appeared exclusively in 'blight' calls and 37% (415) appeared only in 'non-blight' calls. The remaining 17% (182) were found in both blight and non-blight records. Of the 182 tokens that appeared in both 'blight' and 'non-blight' calls, many appeared in roughly the same proportion in both categories, so we chose to restrict our analysis to the 46% of tokens found exclusively in blight-related calls. Using this list of 516 tokens, we calculated the count of unique blight-related terms in each record (mean 7.08, SD 4.4; min/max 0–22) to use as predictor variable for the probability that a call was related to urban blight. The count of unique tokens was a significant predictor of blight-related calls when used as a single predictor (~ 0.37 [0.002], $p < 0.0001$), combined with borough (~ 0.37 [0.002], $p < 0.0001$), and when combined with borough and responsible agency (~ 0.38 [0.002], $p < 0.0001$) (see Table 3 for all coefficient statistics in each model).

**Table 2. Summary of 311 call data.**

| | |
|---|---|
| Total calls, January 1-June 30, 2018 | 1,344,402 |
| Total complaint types | 236 |
| Complaint types with $\geq$ 1,000 complaints ('high frequency') | 93 |
| Percent of complaints in 'high frequency' category | 98.0% |
| Total complaint types considered 'urban blight' | 55 |

**Table 3. Logistic regression model results.**

| Pred(Urban blight = 1) | Model 1 | Model 2 | Model 3 |
|---|---|---|---|
| | Coefficient (SE) | Coefficient (SE) | Coefficient (SE) |
| Intercept | -1.27 (0.009)** | -0.74 (0.014)** | -17.75 (86.15) |
| Unique token count | 0.37 (0.002)** | 0.37 (0.002)** | 0.38 (0.002)** |
| **Borough** (reference = Bronx) | | | |
| Brooklyn | | -0.58 (0.014)** | -0.37 (0.017)** |
| Manhattan | | -0.38 (0.016)** | -0.08 (0.020)** |
| Queens | | -0.76 (0.014)** | -0.58 (0.018)** |
| Staten Island | | -0.21 (0.022)** | 0.13 (0.028)** |
| **Agency** (reference = not specified) | | | |
| Environmental Protection | | | 18.46 (86.15) |
| Department for the Aging | | | -1.85 (168.36) |
| Buildings | | | 17.03 (86.15) |
| Education | | | 32.77 (210.93) |
| Finance | | | -0.40 (95.46) |
| Health and Mental Hygiene | | | 17.03 (86.15) |
| Transportation | | | 21.01 (86.15) |
| Parks and Recreation | | | 19.45 (86.15) |
| Sanitation | | | 15.81 (86.15) |
| Housing Preservation and Development | | | 18.24 (86.15) |
| Human Resources Administration | | | 4.45 (130.11) |
| Police | | | 15.53 (86.15) |
| Taxi and Livery Commission | | | 0.35 (107.37) |

* $p < 0.05$.

** $p < 0.001$.

Model 1: Intercept and unique token count.

Model 2: Intercept, unique token count, and borough.

Model 3: Intercept, unique token count, borough, and agency.

The coefficients for each of the three models were used to predict the probability that a call was blight-related in sample of the data not used in modeling. The predicted values from each model were used to calculate ACC, PPV, and NPV. All models displayed similar PPV levels (90% - 91%); however, model 3 resulted in the best NPV (76%) and ACC (86%) (Table 1).

Model 3 coefficients were then applied to the full data set to calculate and map the predicted percentage of calls that were blight-related by census tract (Fig 1). We found that blight-related calls were concentrated in upper Manhattan, specifically Harlem and the Upper West Side, and the Bronx, which is just north of Manhattan (Example 1A), with some areas of concentration in central Brooklyn—Bedford-Stuyvesant, Crown Heights, Flatbush, and Brownsville (Example 1B). These areas of the city are historically among the most economically distressed, but also observing rapid gentrification. The lowest proportion of calls were observed in the Bay Ridge and Bensonhurst neighborhoods in the southwest of Brooklyn (Example 2A), and in Ridgewood, Middle Village, and Forest Hills in central Queens (Example 2B). These areas in Brooklyn and Queens are highly residential communities with a more pronounced suburban character.

Although the confusion matrices tested internal validity of our models, we next sought to evaluate the construct's external validity by comparing the results to census tract-level housing vacancies from American Community Survey and more current housing and commercial

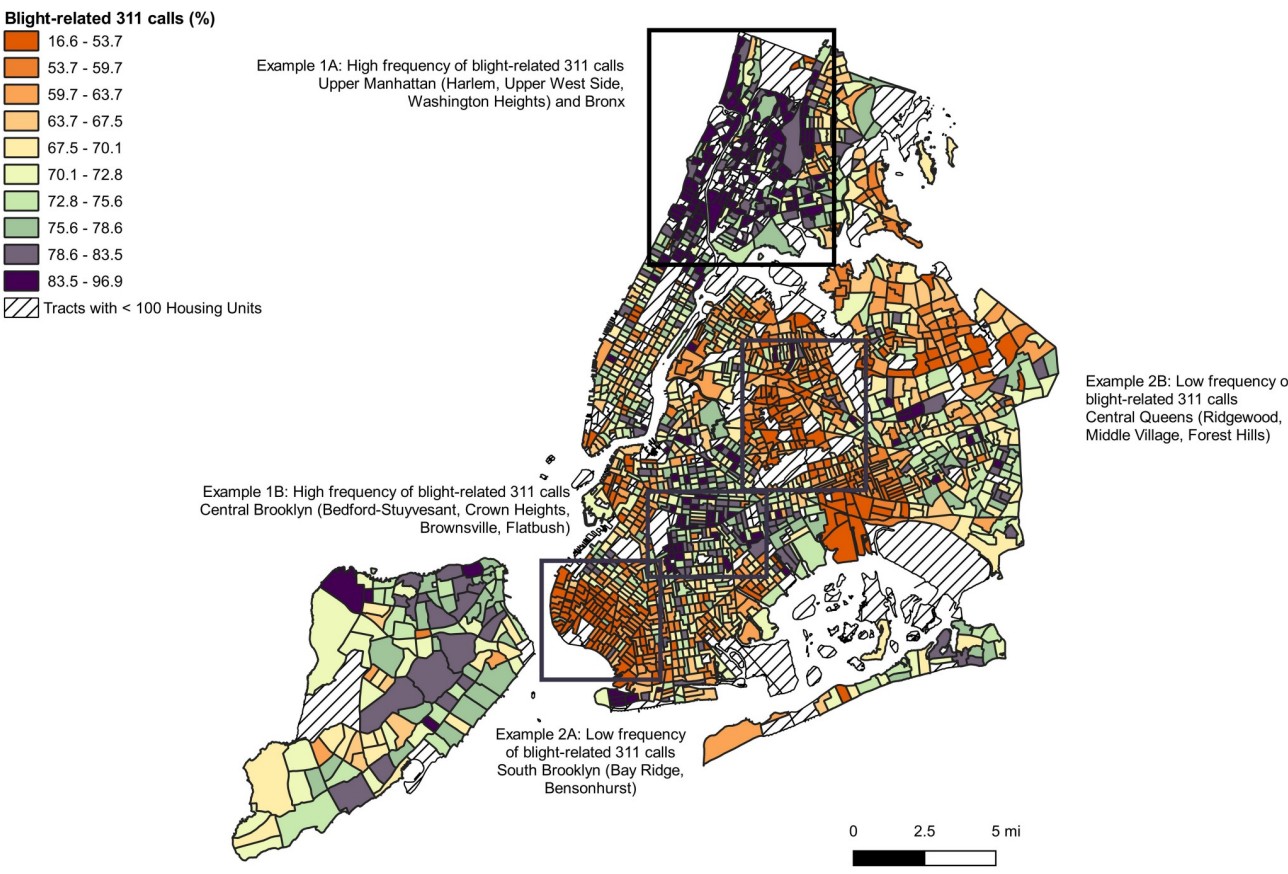

**Fig 1. Map of blight-related 311 calls by census tract.**

vacancy data from USPS/HUD. The correlations between housing vacancies from the American Community Survey and blight-related calls at the tract-level was 0.32 (p < 0.001) for count variables. When expressed as percentages, the correlation was -0.1 (p < 0.001). It seems likely that the positive association between count variables reflects size of the census tracts. When

**Table 4. Correlations between urban blight-related calls and short-, medium-, and long-term vacancies (residential and commercial).**

| Vacancy type | Vacancy Duration | Correlation Coefficient |
|---|---|---|
| Residential | Short-term—<6 mo (%) | 0.026 |
| | Medium-term—6–12 mo (%) | 0.019 |
| | Long-term—>1 yr (%) | 0.098** |
| Commercial | Short-term—<6 mo (%) | 0.049* |
| | Medium-term—6–12 mo (%) | 0.035 |
| | Long-term—>1 yr (%) | 0.16** |
| Total | Short-term—<6 mo (%) | 0.032 |
| | Medium-term—6–12 mo (%) | -0.03 |
| | Long-term—>1 yr (%) | -0.006 |

* p < 0.05.
** p < 0.001.

normalized by total calls and total housing units per census tract, there is a slight, negative relationship between vacancy rates and blight-related calls.

Census tract vacancy data from USPS/HUD presented as percent of residential, commercial, or total vacant addresses showed null or positive associations with percent of blight-related calls. Long-term commercial vacancies had the strongest association with our blight metric, with a correlation of 0.16 (p < 0.0001). Long-term residential vacancies were also associated with blight-related calls, though the correlation was not as strong (0.10, p < 0.0001). Short-term commercial vacancies were also mildly correlated with urban blight-related calls (0.05, p < 0.0001), but none of the remaining short- or medium-term vacancies were statistically significantly correlated with the blight metric (Table 4).

## Discussion

The identified key words were an effective predictor of blight-related calls, but the small, inverse relationship between percent of blight-related calls and vacancy rates based on American Community Survey data was unexpected. The American Community Survey data is based on survey responses over a five-year time frame, so even though it was the most current available, these data preceded the 311 data by 3 years on average. When using HUD/USPS vacancy data for the same time period as the 311 data, disambiguating between residential and commercial vacancies, and specifying duration of vacancies, we found that commercial vacancies —at least in the New York City context—were positively associated with the urban blight metric. The strongest correlation was between long-term (> 12 months) commercial vacancies and percent of calls identified as blight related, which suggests that longer-term vacancies better reflect the 'physical disorder and decay' that we are considering to be urban blight, whereas shorter term vacancies may reflect a certain degree of turnover or 'churning' in the real estate market. Even though the correlation between long-term commercial vacancy and percent of 311 calls related to blight is the strongest association we observed, it is still relatively small. This finding is not necessarily problematic, as we would anticipate each measure to reflect different (unobserved) characteristics of a neighborhood.

Our approach to coding calls as urban blight versus non-urban blight-related relies upon our identification of seven 'domains' of blight which guided our call assignment. The natural extension of this approach is to further develop our algorithm to identify which tokens are predictive of each blight domain. Continuing our algorithm development to predict blight domains will be useful in identifying variations in neighborhoods based on specific components of urban blight. Together with qualitative data collection, these steps can help us determine if areas of blight concentration align with residents' perception of disinvestment in their communities.

These findings are subject to a series of limitations. Most notably, our analysis does not control for a neighborhood's propensity to call 311. As Weaver and Bagchi-Sen note, urban blight can be considered to represent a threshold of 'non-acceptance,' or the point at which community residents find that neighborhood quality has fallen below community-specific norms [8]. These norms are highly variable across neighborhoods, so sidewalk damage on the Upper East Side of Manhattan, which is one of the more affluent communities in New York City, may elicit many more 311 calls relative to a similar condition in Brownsville, Brooklyn, which is a predominantly low-income neighborhood with a high density of public housing. Second, a recent trend in community activation and engagement has emerged in which residents flood 311 dispatch with complaints to motivate city government to repair long-ignored problems in their neighborhoods [11]. Such engagement, while laudable, could make 311-based estimates of urban blight less reliable. If researchers use historical trends to identify propensities for

calling 311, a sudden spike in 311 engagement may erroneously indicate rapid deterioration of the neighborhood's built environment. Finally, a most salient critique of this analysis is how the results may be misused. The findings are not intended as a referendum on residents' interest or willingness to invest in their communities. Placing blame or responsibility on residents without acknowledging that municipal government and the private sector often resist investing in less-affluent, majority-minority neighborhoods only reinforces a cycle of continued disinvestment.

## Conclusion

There is a strong utility for this research amongst urban planners, public health practitioners, and government officials. For urban planners, geographic and temporal patterns in urban blight-related 311 calls (i.e., variation in residents' acceptance of blighted conditions) will help prioritize community needs and desires when determining new planning projects. Such information is essential to develop neighborhoods and amenities that address the most pressing issues communities face.

Moreover, given the connection between blight, biomarkers, and mental health measures, a clearer view of how urban blight is distributed geographically will help public health practitioners identify areas of concentrated poor health, or areas at risk of negative health outcomes across the city. Data on blight-related 311 calls will help public health officials understand where best to concentrate health interventions. For government officials more generally, understanding the key links between urban blight, public health and community investment will help identify where and how cities can maximize the benefits of neighborhood interventions.

The utility of the 311 algorithm will be expanded as it is refined for predicting different domains of urban blight. Although not fully explored here, the category types in the 311 data is a useful lever for distinguishing domains of urban blight. Positive associations with temporally aligned HUD/USPS vacancy data—a commonly used proxy for urban blight—represents a positive step in external validation. As noted, validation will be a continuing process, incorporating domains of urban blight and insights from focus groups drawn from various neighborhoods across New York City. A second step is to assess how predictive our 311 measure of urban blight and its domains are of health-related measures, including but not limited to prevalence of chronic disease, injuries (accidents, interpersonal violence), and mental health conditions. Such associations would provide policy guidance for focusing public health interventions.

## Acknowledgments

We would like to thank Susan Kum, Ph.D., New York City Department of Health and Mental Hygiene for her supporting background research.

## Author Contributions

**Conceptualization:** Jessica Athens, Setu Mehta, Nupur Chaudhury.

**Data curation:** Setu Mehta.

**Formal analysis:** Jessica Athens, Sophie Wheelock.

**Investigation:** Jessica Athens.

**Methodology:** Jessica Athens, Nupur Chaudhury.

**Supervision:** Mark Zezza.

**Validation:** Sophie Wheelock.

**Writing – original draft:** Jessica Athens, Setu Mehta.

**Writing – review & editing:** Jessica Athens, Sophie Wheelock, Nupur Chaudhury, Mark Zezza.

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
