## [Decision Letter · Decision Letter 0]

20 Apr 2020

PONE-D-20-05006

Using 311 data to develop an algorithm to identify urban blight for public health improvement

PLOS ONE

Dear Dr Athens,

Thank you for submitting your manuscript to PLOS ONE. After careful consideration, we feel that it has merit but does not fully meet PLOS ONE’s publication criteria as it currently stands. Therefore, we invite you to submit a revised version of the manuscript that addresses the points raised during the review process.

We would appreciate receiving your revised manuscript by Jun 04 2020 11:59PM. To enhance the reproducibility of your results, we recommend that if applicable you deposit your laboratory protocols in protocols.io, where a protocol can be assigned its own identifier (DOI) such that it can be cited independently in the future. For instructions see: http://journals.plos.org/plosone/s/submission-guidelines#loc-laboratory-protocols

We look forward to receiving your revised manuscript.

Kind regards,

Changshan Wu

Academic Editor

PLOS ONE

3. We note that Figure 2 in your submission contains map images which may be copyrighted. All PLOS content is published under the Creative Commons Attribution License (CC BY 4.0), which means that the manuscript, images, and Supporting Information files will be freely available online, and any third party is permitted to access, download, copy, distribute, and use these materials in any way, even commercially, with proper attribution. For these reasons, we cannot publish previously copyrighted maps or satellite images created using proprietary data, such as Google software (Google Maps, Street View, and Earth). For more information, see our copyright guidelines: http://journals.plos.org/plosone/s/licenses-and-copyright.

1.    You may seek permission from the original copyright holder of Figure 2 to publish the content specifically under the CC BY 4.0 license. 

Reviewers' comments:

Reviewer's Responses to Questions

**Comments to the Author**

1. Is the manuscript technically sound, and do the data support the conclusions?

Reviewer #1: Partly

Reviewer #2: Yes

2. Has the statistical analysis been performed appropriately and rigorously? 

Reviewer #1: Yes

Reviewer #2: Yes

3. Have the authors made all data underlying the findings in their manuscript fully available?

Reviewer #1: No

Reviewer #2: Yes

4. Is the manuscript presented in an intelligible fashion and written in standard English?

Reviewer #1: Yes

Reviewer #2: Yes

5. Review Comments to the Author

Reviewer #1: The authors used natural language processing to categorize 311 data from NYC into different blight-related categories, and constructed models to identify blight-related calls using keywords/tokens. In my opinion, the manuscript presents an interesting and promising use of municipal open data to better understand how residents perceive and report blight in a city. However, the manuscript needs to be improved in several areas before it should be considered for publication.

First, the ACS vacancy data were analyzed as both raw counts and as percentages, and the authors found that the percentage data are likely more appropriate because the count data largely reflect the size of the census tract (p13). Given this strong rationale for analyzing data as percentages, why was the USPS/HUD data analyzed as counts but not as a percentage? The USPS/HUD data set provides total residential/business addresses in each tract, so this analysis is feasible and should be added unless there is a compelling reason not to do so.

Writing:

There is room to improve the writing and organization throughout the manuscript, but particularly in the Introduction. This will help the reader better understand the motivations and implications of the study. Some primary suggestions are as follows:

* Better transitions within and between sentences and paragraphs would be helpful. For example, on p3 para1, the colon in “…social ills: Residential overcrowding…” is not appropriate and the R should not be capitalized. On p4 para2, the sentence would read smoother as “…are highly problematic, because they rely…” On p4 para3, the colon after numerous does not work there, and I suggest placing a period after numerous to end the sentence.

* More strategic organization of text. For example, on p3 para2 the first sentence is out of order chronologically with the second sentence, and it actually seems to me like the first sentence should be dropped or moved because it doesn’t fit very well with the rest of the paragraph.

* Be more specific where possible. On p3 para 2, both ‘many aspects’ and ‘the facets of’ are vague and could be described more specifically. Other examples from the Discussion are noted below.

* Per journal instructions, please include line numbers. This makes it much easier to suggest edits to the manuscript.

Line item comments:

Abstract:

* p<0.001 instead of >

* ‘different facets’: Which facets? Can you be more specific here, or offer a couple examples?

Introduction:

* p3 para1: How can a direct quote have two sources? Cite the original source only.

* p3 para3: (physical decay) could be incorporated more smoothly into the sentence to avoid parentheses. Same with (physical) disorder on the next page.

* p4 para3: ‘affects’ not ‘affect.’

* p4 para3: References are needed after ‘well documented’, and in the following sentence. In the sentence after that, please place each reference after the specific outcome it addresses rather than grouping the references at the end.

* p5 para3: Maybe ‘validated’ is the best word here, but I wonder if ‘compared to’ isn’t a more appropriate verb. I understand model validation to be the process of making sure the model performs as expected; here, the authors use the ACS & USPS data sets to draw inferences about the relationships between blight-related calls and vacancy rates rather than using ACS & USPS data to characterize the quality of the algorithm itself. To me, this seems less like model validation and more like a statistical comparison in which the algorithm results are assumed to be valid.

Methods:

* This section is much clearer than the Introduction.

Results:

* ‘46% of token’ should be tokens

* p12 para1: missing ‘the’ in first sentence

* p12 para1: Table 1, not Table 4

* p12 para 2: Noting specific locations within NYC is only useful to those familiar with NYC because these locations are not shown on the map figure

* p13 and elsewhere: Inconsistent use of ACS, American Community Survey, and Census. Please define the abbreviation at first use and then use it consistently thereafter.

* p13 para1: correlation, not correlations

* p13 para1: why is count italicized?

Discussion:

* p14: The paragraph beginning ‘Nevertheless’ is not a complete paragraph.

* p14: change to ‘further develop our algorithm’

* p15 para1: Why would calls differ between these two places? I am not familiar with these places, so please explain why we should expect differences.

* p15 para1: Please add an explanation for why flooding the 311 lines would make estimates of blight less reliable.

* p15 para1: Please explain how blight on vacant commercial properties (which I take to be an important finding based on earlier discussion material) shows disinvestment by municipal government. And why is municipal government italicized here?

Table 1: Please provide a more descriptive caption so the table can stand alone

Fig 1: Please explain the axis ranges here. How can a word frequency go below 0%?

Fig 2: This is not a density map, so the caption should be reworded to more accurately describe the contents. In addition, the map really ought to show & label the borough outlines because boroughs are discussed in the text. As someone who is largely unfamiliar with NYC, labeling places within boroughs that are discussed in the text would be helpful for context (JFK, Brownsville, etc.).

Reviewer #2: This manuscript attempted to identify urban blights using 311 data. Particularly, several major types of "blights" were identified, and regression analysis was performed to examine the effectiveness of the model. This is a well-written and straightforward paper with contributions to the literature. My major comments are as follows.

1) Methodology: this paper does not provide a detailed algorithm for extracting the "blight" information from natural languages. As the authors claimed that one major contribution is to develop a new algorithm. The authors have specify the algorithm, and highlight their contributions.

2) Conclusions: this paper does not have a conclusion session. The authors need summarize their major results of the paper.

6. PLOS authors have the option to publish the peer review history of their article (what does this mean?). If published, this will include your full peer review and any attached files.

Reviewer #1: No

Reviewer #2: No

---

## [Author Response · Author response to Decision Letter 0]

4 Jun 2020

Response to Reviewers

Comments from Editor

1. Please ensure that your manuscript meets PLOS ONE's style requirements, including those for file naming. We confirmed that formatting met PLOS ONE’s style requirements.

2. PLOS requires an ORCID iD for the corresponding author in Editorial Manager on papers submitted after December 6th, 2016. Please ensure that you have an ORCID iD and that it is validated in Editorial Manager. We included the ORCID ID for the corresponding author.

3. We require you to either (1) present written permission from the copyright holder to publish these figures specifically under the CC BY 4.0 license, or (2) remove the figures from your submission. We modified the figures so that we are not using any copyrighted materials.

4. The PLOS Data policy requires authors to make all data underlying the findings described in their manuscript fully available without restriction, with rare exception (please refer to the Data Availability Statement in the manuscript PDF file). The data should be provided as part of the manuscript or its supporting information, or deposited to a public repository. R script for calling the underlying 311 data is provided as an appendix to the manuscript.

Global Comments from Reviewer 1

• Writing: Better transitions within and between sentences and paragraphs would be helpful. For example, on p3 para1, the colon in “…social ills: Residential overcrowding…” is not appropriate and the R should not be capitalized. On p4 para2, the sentence would read smoother as “…are highly problematic, because they rely…” On p4 para3, the colon after numerous does not work there, and I suggest placing a period after numerous to end the sentence. Thank you to reviewer 1 for your careful review and edits. We attempted to address all of your comments, particularly those related to manuscript organization, transitions between topics, and specificity in examples. I believe though that the manuscript is in much better shape due to the changes. 

Specific edits are outlined below 

• More strategic organization of text. For example, on p3 para2 the first sentence is out of order chronologically with the second sentence, and it actually seems to me like the first sentence should be dropped or moved because it doesn’t fit very well with the rest of the paragraph. Specific edits are outlined below. 

Comments on Introduction

• Be more specific where possible. On p3 para 2, both ‘many aspects’ and ‘the facets of’ are vague and could be described more specifically. Other examples from the Discussion are noted below We added further details on page 2, lines 28-32. 

• Per journal instructions, please include line numbers. This makes it much easier to suggest edits to the manuscript. Line numbers have been added to the manuscript.

p<0.001 instead of >

‘different facets’: Which facets? Can you be more specific here, or offer a couple examples? The > sign has been correct. ‘Different facets’ have been further explicated in lines 28-32: ‘These facets include physical disorder (e.g., litter, overgrown lawns, or graffiti) and decay (e.g., vacant or abandoned lots or sidewalks in disrepair) that are manifestations of social processes such as (loss of) neighborhood cohesion, social control, collective efficacy, and anchor institutions.’

• p3 para1: How can a direct quote have two sources? Cite the original source only. The citations were corrected so that the quote has a single, correct citation. 

• p3 para3: (physical decay) could be incorporated more smoothly into the sentence to avoid parentheses. Same with (physical) disorder on the next page. These paragraphs were edited to remove the use of the parenthetical. 

“Standard approaches to measuring blight with secondary data sets include measures of physical decay, such as vacant commercial property, vacant housing units, and vacant lots within a neighborhood [7,8]. Other researchers have looked at measures of physical disorder, otherwise considered neighborhood quality measures, such as mown lawns, litter and debris, delinquent vehicles, and presence/absence of graffiti [7,9]. These quality-related indicators are more difficult to capture, often requiring street audits in person or virtually, using Google Street View [10,11]. Finally, given that blight is considered a physical manifestation of poor social cohesion, a third dimension of blight includes social and economic investment, which can be evaluated through measures such as perceptions of safety, presence of anchor institutions, and community organizing efforts [12].”

• p4 para3: ‘affects’ not ‘affect.’ Made this edit. 

• p4 para3: References are needed after ‘well documented’, and in the following sentence. In the sentence after that, please place each reference after the specific outcome it addresses rather than grouping the references at the end. All of these citations are review papers that address all or most of the aspects referred to in the sentence. However, citations are broken out later in the paragraph.

“Though the literature specific to urban blight and public health outcomes is less developed, there is strong evidence that urban blight is specifically associated with higher violent crime and gun crime [6], poor mental health [18-19], and even adverse pregnancy outcomes [20].”

• p5 para3: Maybe ‘validated’ is the best word here, but I wonder if ‘compared to’ isn’t a more appropriate verb. I understand model validation to be the process of making sure the model performs as expected; here, the authors use the ACS & USPS data sets to draw inferences about the relationships between blight-related calls and vacancy rates rather than using ACS & USPS data to characterize the quality of the algorithm itself. To me, this seems less like model validation and more like a statistical comparison in which the algorithm results are assumed to be valid. Thank you for this point – we made this edit. 

Comments on Results:

• ‘46% of token’ should be tokens Made this correction.

• p12 para1: missing ‘the’ in first sentence Made this correction.

• p12 para1: Table 1, not Table 4 Made this correction.

• p12 para 2: Noting specific locations within NYC is only useful to those familiar with NYC because these locations are not shown on the map figure Specific communities are now identified on the map and in the text.

• p13 and elsewhere: Inconsistent use of ACS, American Community Survey, and Census. Please define the abbreviation at first use and then use it consistently thereafter. These corrections have been made.

• p13 para1: correlation, not correlations This correction has been made.

• p13 para1: why is count italicized? This has been set to normal font

Comments on Discussion:

• p14: The paragraph beginning ‘Nevertheless’ is not a complete paragraph. This sentence was combined with the previous paragraph. 

• p14: change to ‘further develop our algorithm’ This change was made.

• P15 para1: Why would calls differ between these two places? I am not familiar with these places, so please explain why we should expect differences. Additional details were included to explain why these communities may differ in their propensity to call 311.

“These findings are subject to a series of limitations. Most notably, our analysis does not control for a neighborhood’s propensity to call 311. As Weaver and Bagchi-Sen note, urban blight can be considered to represent a threshold of ‘non-acceptance,’ or the point at which community residents find that neighborhood quality has fallen below community-specific norms [9]. These norms are highly variable across neighborhoods, so sidewalk damage on the Upper East Side of Manhattan, which is one of the more affluent communities in New York City, may elicit many more 311 calls relative to a similar condition in Brownsville, Brooklyn, which is a predominantly low-income neighborhood with a high density of public housing.”

• p15 para1: Please add an explanation for why flooding the 311 lines would make estimates of blight less reliable The text has been revised to better address this question.

“Such engagement, while laudable, could make 311-based estimates of urban blight less reliable. If researchers use historical trends to identify propensities for calling 311, a sudden spike in 311 engagement may erroneously indicate rapid deterioration of the neighborhood’s built environment.”

• p15 para1: Please explain how blight on vacant commercial properties (which I take to be an important finding based on earlier discussion material) shows disinvestment by municipal government. And why is municipal government italicized here? This text has been updated to better represent the pattern of disinvestment. Italics have been removed from ‘municipal’

“Placing blame or responsibility on residents without acknowledging that municipal government and the private sector often resist investing in less-affluent, majority-minority neighborhoods only reinforces a cycle of continued disinvestment.” 

Comments on Tables and Figures

• Table 1: Please provide a more descriptive caption so the table can stand alone The caption has been changed to “Table 1. Sensitivity, specificity, and accuracy of urban blight algorithm.” 

• Fig 1: Please explain the axis ranges here. How can a word frequency go below 0%? This figure has been removed because it has limited added value to the paper and its findings. The text has been modified to the following:

“The remaining 17% (182) were found in both blight and non-blight records. Of the 182 tokens that appeared in both ‘blight’ and ‘non-blight’ calls, many appeared in roughly the same proportion in both categories, so we chose to restrict our analysis to the 46% of tokens found exclusively in blight-related calls.” 

• Fig 2: This is not a density map, so the caption should be reworded to more accurately describe the contents. In addition, the map really ought to show & label the borough outlines because boroughs are discussed in the text. As someone who is largely unfamiliar with NYC, labeling places within boroughs that are discussed in the text would be helpful for context (JFK, Brownsville, etc.). The map has been relabeled as a choropleth map, and areas of the city that are discussed in the text are indicated on the map as well.

Global Comments from Reviewer 2

• Methodology: this paper does not provide a detailed algorithm for extracting the "blight" information from natural languages. As the authors claimed that one major contribution is to develop a new algorithm. The authors have specify the algorithm, and highlight their contributions We described our strategy for identifying blight information from 311 text in the following text:

“Training the data requires the manual designation of a random sample of calls to an urban blight versus a non-urban blight category. We established seven domains of urban blight based on extant literature to guide our data training: social conditions, abandoned property, air quality, street/sidewalk maintenance, noise, sanitary conditions, and building safety. Coding any call into one of these categories signifies the call is urban blight-related. Domains were assigned based on complaint types and free-text call descriptions in the 311 data. Complaint type is a variable designated by the City of New York and included as part of the 311 system data, which comprises 236 complaint types. We focused on ‘high frequency’ complaints (≥ 1,000 records) to simplify the training process. The high frequency list included 93 complaint types.”

• Conclusions: this paper does not have a conclusion session. The authors need summarize their major results of the paper. The Discussion section was expanded and used to create a “Conclusions” section, which starts at line 308.

---

## [Decision Letter · Decision Letter 1]

11 Jun 2020

Using 311 data to develop an algorithm to identify urban blight for public health improvement

PONE-D-20-05006R1

Dear Dr. Athens,

We’re pleased to inform you that your manuscript has been judged scientifically suitable for publication and will be formally accepted for publication once it meets all outstanding technical requirements.

Kind regards,

Changshan Wu

Academic Editor

PLOS ONE

Additional Editor Comments (optional):

Reviewers' comments:

Reviewer's Responses to Questions

**Comments to the Author**

1. If the authors have adequately addressed your comments raised in a previous round of review and you feel that this manuscript is now acceptable for publication, you may indicate that here to bypass the “Comments to the Author” section, enter your conflict of interest statement in the “Confidential to Editor” section, and submit your "Accept" recommendation.

Reviewer #1: (No Response)

Reviewer #2: All comments have been addressed

2. Is the manuscript technically sound, and do the data support the conclusions?

Reviewer #1: Yes

Reviewer #2: Yes

3. Has the statistical analysis been performed appropriately and rigorously? 

Reviewer #1: Yes

Reviewer #2: Yes

4. Have the authors made all data underlying the findings in their manuscript fully available?

Reviewer #1: Yes

Reviewer #2: Yes

5. Is the manuscript presented in an intelligible fashion and written in standard English?

Reviewer #1: Yes

Reviewer #2: Yes

6. Review Comments to the Author

Reviewer #1: One very minor remaining issue: update figure numbers in the text, because figure 1 was deleted.

Also, consider changing the color ramp on the map. It seems strange to have low percentages of blight shown as red, as red is typically used to connote warning and draw the eye.

Reviewer #2: The authors have successfully addressed my comments. I am happy to see this manuscript to be accepted for publication.

7. PLOS authors have the option to publish the peer review history of their article (what does this mean?). If published, this will include your full peer review and any attached files.

Reviewer #1: No

Reviewer #2: No

---

## [Editor Report · Acceptance letter]

25 Jun 2020

PONE-D-20-05006R1 

Using 311 data to develop an algorithm to identify urban blight for public health improvement 

Dear Dr. Athens:

I'm pleased to inform you that your manuscript has been deemed suitable for publication in PLOS ONE. Congratulations! Your manuscript is now with our production department. 

Kind regards, 

on behalf of

Dr. Changshan Wu 

Academic Editor

PLOS ONE